# Time Series Analysis of Water Quality Factors Enhancing Harmful Algal Blooms (HABs): A Study Integrating In-Situ and Satellite Data, Vaal Dam, South Africa

Altayeb A. Obaid [1,2,*], Elhadi M. Adam [1], K. Adem Ali [1,3] and Tamiru A. Abiye [4]

1  School of Geography, Archaeology and Environmental Studies, University of the Witwatersrand, Private Bag X3, P.O. Box Wits, Johannesburg 2050, South Africa; elhadi.adam@wits.ac.za (E.M.A.); alika@cofc.edu (K.A.A.)
2  Faculty of Environmental Sciences & Natural Resources, University of Al-Fashir, Al Fashir P.O. Box 125, Sudan
3  Department of Geology and Environmental Geosciences, College of Charleston, Charleston, SC 29424, USA
4  School of Geosciences, University of the Witwatersrand, Private Bag X3, P.O. Box Wits, Johannesburg 2050, South Africa; tamiru.abiye@wits.ac.za
*  Correspondence: 2126105@students.wits.ac.za

**Abstract:** The Vaal Dam catchment, which is the source of potable water for Gauteng province, is characterized by diverse human activities, and the dam encounters significant nutrient input from multiple sources within its catchment. As a result, there has been a rise in Harmful Algal Blooms (HABs) within the reservoir of the dam. In this study, we employed time series analysis on nutrient data to explore the relationship between HABs, using chlorophyll-a (Chl−a) as a proxy, and nutrient levels. Additionally, Chl−a data extracted from Landsat-8 satellite images was utilized to visualize the spatial distribution of HABs in the reservoir. Our findings revealed that HAB productivity in the Vaal Dam is influenced by the levels of total phosphorus (TP) and organic nitrogen (KJEL_N), which exhibited a positive correlation with chlorophyll-a (Chl−a) concentration. Long-term analysis of Chl−a in-situ data (1986–2022) collected at a specific point within the reservoir showed an average concentration of 11.25 μg/L. However, on certain stochastic dates, Chl−a concentration spiked to very high values, reaching a maximum of 452.8 μg/L, coinciding with elevated records of TP and KJEL_N concentrations on those dates, indicating their effect on productivity levels. The decadal time series and trend analysis demonstrated an increasing trend in Chl−a productivity over the studied period, rising from 4.75 μg/L in the first decade (1990–2000) to 10.51 μg/L in the second decade (2000–2010), and reaching 16.7 μg/L in the last decade (2010–2020). The rising averages of the decadal values were associated with increasing decadal averages of its driving factors, TP from 0.1043 to 0.1096 to 0.1119 mg/L for the three decades, respectively, and KJEL_N from 0.80 mg/L in the first decade to 1.14 mg/L in the last decade. Satellite data analysis during the last decade revealed that the spatial dynamics of HABs are influenced by the dam's geometry and the levels of discharge from its two feeding rivers, with higher concentrations observed in meandering areas of the reservoir and within zones of restricted water circulation.

**Keywords:** Vaal Dam; time series; chlorophyll-a; harmful algal blooms

## 1. Introduction

Globally, freshwater resources have been put under increasing pressure due to the rapid increase in the world population and their improvement of living standards [1]. It is indicated that an estimation of more than 83% of land surfaces that are surrounding freshwater systems have been significantly influenced by the footprint of human beings as a response to anthropogenic activities [1,2]. Contaminants from ineffective waste management, pesticides and fertilizers from agricultural areas, pollution from urban, industrial,

and domestic wastewater can often be released to ground and surface water [3], and even the landfills or slag heap disposals may release pollutants seeping into nearby water resources [4,5]. The inland freshwater bodies are more vulnerable to such problems of pollution and contamination [6] resulting in the decline of quality and availability. Among the loaded pollutants, some serve as nutrients for the enhancing growth of algae. Such nutrients can be loaded from point or non-point sources, point sources, for example, wastewater treatment plants can easily be recognized and given more attention [7,8]. Non-point sources including urban areas, cultivated lands, natural forests, and pastures are more difficult to recognize and attention should be given to them according to the amounts of nutrients they can potentially release. For example, it was found that the estimated loads of nitrogen, phosphorus, and suspended sediments from urban areas and cultivated land are 10 to 100 times greater than idle and forested lands in the Great Lakes catchments of USA and Canada [9].

Eutrophication is a serious freshwater problem caused by the excessive growth of harmful algal blooms (HABs). In recent years, the factors enhancing such blooms have become interesting research subjects. Many studies have been conducted in this field, and most of them were related to the HABs to the loading of nutrients such as nitrogen and phosphorus into the water bodies [10,11].

In Africa, societies are highly dependent on their natural inland freshwater resources creating more pressure on them resulting in significant changes in their water quality [12]. South Africa has over 4000 freshwater storage dams, among which around 700 are public dams controlled and managed by the governments used for domestic, irrigation, and industrial water supply [13], they are underpinning the economic and social development of the country. Despite such a huge number of dams, South Africa is a water-stressed and scarce country facing critical challenges due to poor water management practices, inadequate infrastructure, and a relentless surge in water demands [14]. Owing to the fact that most South African dams are located downstream of metropolitan and urban areas, they have become more enriched with nutrients [4,15]. The climatic conditions of South Africa associated with nutrient loads resulted in extensive and widespread eutrophication and cyanobacterial blooms in the inland freshwater bodies [16,17]. This will continue to increase the cost of using these valuable resources. Nitrogen and phosphorus are considered the leading factors in accelerating the lakes' and reservoirs' eutrophication [18]. The Water Act of 1956, Section 21(1) (a) and the Department of Water Affairs in 1980 set exceptional standards of the phosphate concentration for effluent discharged to some mentioned rivers or their tributaries, it promulgated that the phosphate concentration should not be higher than 1 mg/L [14,19]. An investigation was conducted by CSIR and the Department of Water Affairs from 1985 to 1988 to predict the impact of the phosphorus standard (1 mg/L). Based on their findings, the eutrophication control aim was set to maintain the mean concentration of chlorophyll-a within receiving water bodies at the level that no conditions of severe nuisance would occur more than 20% of the time. To achieve this aim, it is required to keep the phosphorus concentration level in the reservoirs water less than 0.14 mg/L [20].

Vaal Dam is one of multiple water bodies affected by eutrophication and cyanobacterial blooms in South Africa. It was constructed by the Department of Irrigation of the national government and functionally completed in October 1936 at 20 km downstream of the confluence of the Vaal and Wilge rivers [21]. It is the second biggest dam by surface area in South Africa (about 320 Km$^2$). The dam has undergone several rises to increase its storage capacity which ended up with a total capacity of approximately $2.603 \times 10^6$ m$^3$ [21]. The dam catchment area is approximately 38,000 km$^2$ holding various human activities including major agricultural activities (crop cultivation and cattle grazing), mining, and some industrial activities [22,23] as well as many formal and informal settlements. In many areas within the Vaal Dam catchment, mine dewatering and urban/industrial treated effluent discharges find their way to the streams causing serious water quality issues [22]. Such activities have direct or indirect effects on the dam water quality in terms of nutrient loading which enhances the growth of HABs. A study in 2013 stated that the Vaal Dam

Reservoir was classified as a mesotrophic water body according to the South African Department of Water Affairs Classification System, the mean concentration of chlorophyll-a was 14.8 µg/L, the mean total phosphorus concentration was 0.077 mg/L, and the time percentage where chlorophyll-a exceeds 30 µg/L was found was 17% [24].

HABs have major effects on water quality and their aquatic system function; therefore, monitoring of their distribution in space and time is very important for water resources managers to address the issues related to it. Moreover, it is very important to address the question of which factors enhance and control the HABs in the Vaal Dam Reservoir. Historically, many studies were conducted to assess different methods to detect HABs and cyanobacteria in the Vaal Dam Reservoir, for example, an attempt was made to help the managers of the drinking water treatment facility with advanced prediction of *Microcystis* sp. concentration in the Vaal Dam water, by building a model using physical, chemical, and biological water quality records between 2000 and 2012. The model showed a promising result in estimating *Microcystis* sp. in 7 days in advance [24]. The composition of the phytoplankton in the reservoir was examined over the course of a year, it was found that Anabaena and Microcystis are the most abundant cyanobacteria that dominate the phytoplankton on the dam reservoir during summer time; during February and March, the study reviled that they are not necessary dominating throughout the year [25]. Few studies were conducted in the Vaal Dam using remote sensing data (Landsat 8 OLI and Sentinel 2 MSI), the two sensors have relatively high spatial resolution and a capability to distinguish very small differences in the optics of the water column as a function of water quality parameters such as Chl−a. Sakuno, Y et al., (2018) have obtained a strong correlation between the satellite estimation using unified red-to-near-infrared two-band and three-band models, and the in-situ measurements (see ref. [26]). Another study tested the red-to-near-infrared (NIR-red) bands by means of stepwise logistic regression (SLR) to classify Chl−a concentrations using Landsat 8 data for 2014–2016, the SLR gave overall accuracy ranged between 65 to 80% (see ref. [27]). A comparison of Landsat 8 and Sentinel 2 was conducted in mapping water quality at Vaal Dam, Sentinel 2 generated better results than Landsat 8 for estimating both chlorophyll-a and turbidity with $R^2$ = 0.86 comparing to 0.68 for Chl−a, and 0.59 and 0.50 for turbidity [26–28]. The latest remote sensing-based article published on the Vaal Dam was (Obaid et al., 2021) applied to high-resolution sensors that successfully used both the blue–green and red-infrared OC algorithms in estimating Chl−a concentrations [15,29].

Blue–green algorithms are typically utilized to estimate chlorophyll-a concentration in Case I waters, where water quality is predominantly influenced by phytoplankton [30,31]. However, in Case II waters, where the optical signature is influenced by various water quality constituents, employing blue–green models can result in inaccuracies when estimating chlorophyll-a concentration. The successful use of the blue–green OC algorithm in the Vaal Dam, a Case II water body, may be attributed to the strong signal of HABs arising from the high biomass concentration in the reservoir water column. While previous studies have focused on developing methods to detect HABs and cyanobacterial blooms in the reservoir, none have attempted to uncover the factors contributing to such blooms in terms of nutrient loading and environmental factors.

This study aims to explore the correlation between HABs, nutrients, and environmental factors during peak blooms over recent decades. Historical water quality records from the dam reservoir will be analyzed to investigate this relationship by comparing nutrient levels with measured chlorophyll-a (Chl−a) concentrations, thereby identifying which nutrients contribute to algal bloom enhancement. Additionally, the study seeks to assess the spatial distribution of HABs using Landsat-8 satellite images captured in late summer (April) between 2013 and 2023. This analysis will facilitate the identification of the frequently and most affected areas of the dam reservoir. Leveraging remote sensing techniques provides a synoptic view of HABs' spatial and temporal distribution, enhancing our understanding of their dynamic variations. This approach has the potential to enable effective monitoring of HABs, utilizing high-resolution data for more precise studies.

## 2. The Study Area

The Vaal Dam holds water used to supply potable water to the Gauteng metropolitans and its surrounding areas. The Vaal Dam Catchment extends within Free State, Mpumalanga, and Gauteng provinces and drains by the Vaal and Wilge river systems, (Figure 1). The main rivers consist of many tributaries draining different areas in terms of land use land cover types and human activities.

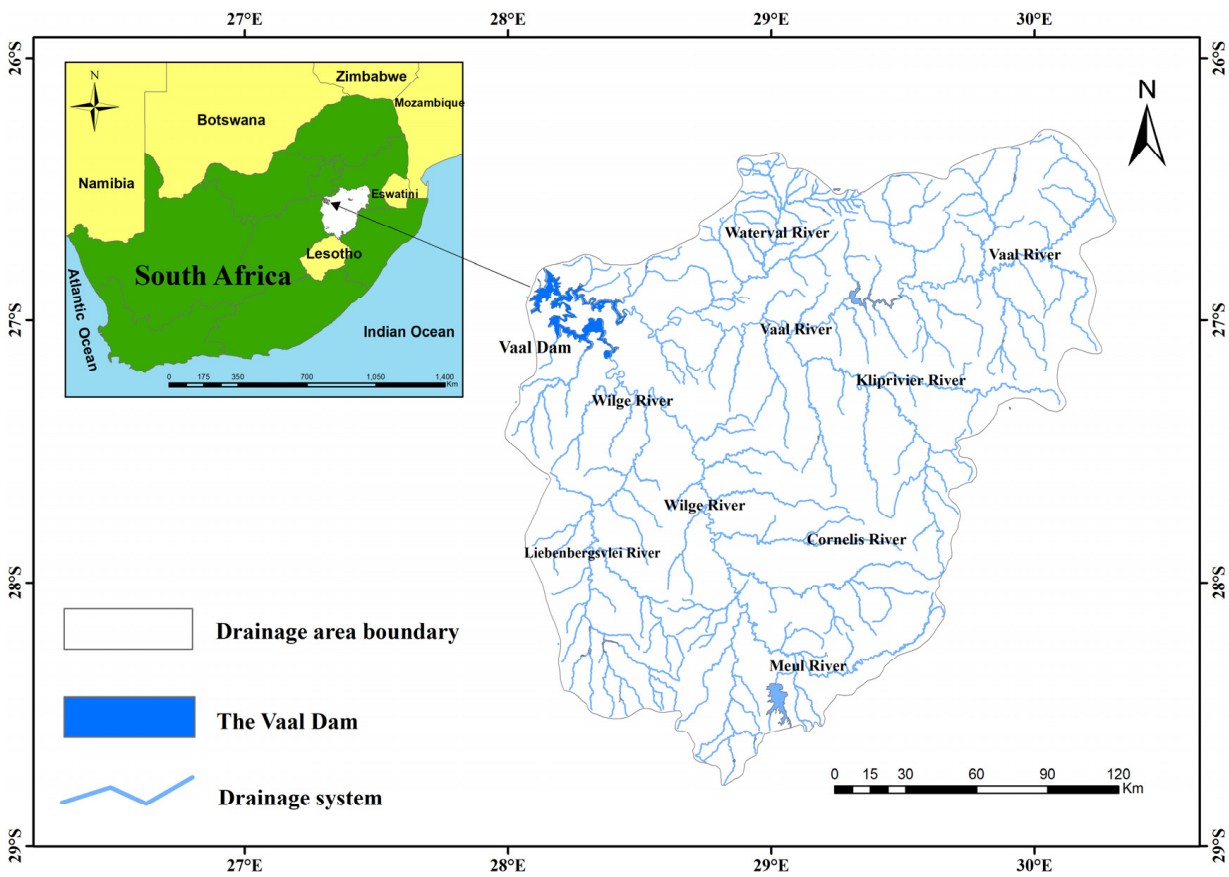

**Figure 1.** The location map of the Vaal Dam.

The Waterval River contributes approximately $111 \times 10^6$ m$^3$ of water annually to the Vaal River. It drains very active areas holding intense human activities such as agriculture, industry, mining, and urban and rural settlements, mainly in the upper reach of the Waterval River, these activities have been shown to be responsible for the deterioration of the water quality and ecological integrity of the river system [30]. Another active area with mining and industrial activities within the Vaal River site is the Grootdraai Dam catchment, it is found that the greatest area of concern was the region's closeness to the downstream of the urban, industrial, mining, and cultivated lands cover [31]. These two active areas in terms of land use activities put the Vaal River under more concern about its water quality issues.

The Wilge River system drains areas dominated by agriculture and grasslands, it contributes a great deal of water to the dam from its catchment and from the Lesotho Highland Water Project [32]. The Tugela–Vaal Water Project also contributes a good portion of water to the Vaal Dam via the Wilge and Nuwejaarspruit Rivers [21].

## 3. Materials and Methods

### 3.1. The Water Quality Data

Historical water quality was obtained from the Department of Water and Sanitation (http://www.dwa.gov.za/iwqs/wms/data/WMA08_reg_WMS_nobor.htm), accessed on

2 January 2023. The data contains chemical and physicochemical water quality parameters from the Vaal Dam Reservoir. Daily water quality data include Chl−a, Total Phosphorus (TP), Dissolved Oxygen (DO), KJEL Nitrogen (KJEL_N), Ammonia Nitrogen (NH$_4$_N), Nitrate and Nitrite Nitrogen (NO$_3$NO$_2$_N), and water Temperature (Temp) have been downloaded. Some gaps of data missing data existed in some chosen parameters, the records were poor with a big gap noticed between 2000 and 2010 for all-targeted water quality parameters except Chl−a and TP. A good record was only found for the period between 2010 to 2019. Thus, the distribution, time series, and decadal trend plots as well as regressions between the variables were generated using the available data only, no gap-fill approach was applied. The continuous DO records are only available from 2010 to 2019.

### 3.2. The Satellite Data

Cloud-free Landsat-8 OLI, Level 1 data were downloaded from the United States Geological Survey (USGS) Earth Explorer website (https://earthexplorer.usgs.gov/), accessed on 1 March 2023. One image was acquired for each summer season in a year, and a total of eleven images were downloaded covering the period between 2013 and 2023 to understand the spatial dynamics of HABs in the reservoir, specifically, the images acquired in April if available otherwise any summer month. The images were processed using a level-2 generator (l2gen) of the NASA SeaDAS software version 8.3, which is the standard ocean color processing software of NASA. To perform the atmospheric correction, l2gen uses an iterative process to estimate the aerosol radiance, it uses the near-infrared and short-wave infrared to overcome the limitation of dark pixel assumption, and it has proven to give a better result in turbid waters [33]. Landsat 8 OLI bands 865 and 2201 were used in this study for atmospheric correction and then normalized water-leaving radiance (*nLw*) and remote sensing reflectance (*Rrs*) have been retrieved in l2gen. All necessary masks have been applied using SeaDAS v.8.3 standard masking to retrieve only the open water body images.

### 3.3. Data Analysis

Historical in-situ water quality data has been analyzed using statistical methods in R v4.3. The long-term average and the standard deviation (SD) for any single variable have been calculated from the whole data. The SD values have been corrected using a correction factor to obtain the known SD in textbooks. Multiplot types including frequency scatter plots, time series plots, and decadal trend plots have been generated. The variable data have been indexed and matched using the index and match function in an Excel sheet. Then, the above-mentioned plots have been generated. The erratic high values have been excluded from the frequency distribution plots to explore the most frequent values while all data have been included in long-term time series (1986 to 2022), decadal time series, and decadal trend plots. The analysis was performed to understand the changes in the studied parameters. The correlation between these variables has also been evaluated, scatter plots of Chl−a against temperature and the potential nutrients, and between the variables that are well-correlated with Chl−a were plotted to reveal the relationship between such variables. From the perusal of the graph, the extremely high values that show excessive HAB conditions were chosen for detailed analysis alongside their concurrent nutrient concentrations. Since the data were collected over only one point within this large dam reservoir, more understanding of the spatial distribution of HABs was conducted using satellite remote sensing data to explore the spatial change of productivity levels. A simple two-band ratio of blue–green based ocean color algorithm was applied [34] for Vaal Dam water modeling productivity. Level 2 Landsat-8 products obtained from level−2 generator (l2gen) of NASA SeaDAS v8.3 were used for chlorophyll-a retrieval. The ratio of (*Rrs*560/*Rrs*443) from our previous work [29] is used in this study, it was successfully used to retrieve Chl−a from Landsat-8 data in the Vaal Dam.

## 4. Results

*Time Series of Targeted Parameters*

Time series of the Chl−a, TP, DO, KJEL_N, NH$_4$_N, NO$_3$NO$_2$_N, and temperature have been plotted from 1986 to 2022. The analysis of the data shows that the concentration levels for the targeted water quality parameters in the Vaal Dam ranged from very low values to extremely high recorded values on some stochastic dates. Table 1 summarizes the availability of the data coverage periods and their minimum and maximum recorded values as well as their averages which are calculated from the whole available data. The averages tend to lower values which explains that the extremely high values do not always occur.

**Table 1.** Summary of the availability periods and the characteristics of the targeted WQ parameters.

| Parameter | Data Availability | Minimum Value | Maximum Value | Average | Standard Deviation |
|---|---|---|---|---|---|
| Chl−a | 1986–2022 | 0.5 µg/L | 452.8 µg/L | 11.25 µg/L | 34.72 |
| TP | 1986–2022 | 0.01 mg/L | 1.4 mg/L | 0.113 mg/L | 0.121 |
| DO | 2010–2018 | 4.98 mg/L | 19.31 mg/L | 7.9 mg/L | 1.615 |
| KJEL_N | 1986–2018 | 0.05 mg/L | 18.058 mg/L | 0.94 mg/L | 1.352 |
| NH$_4$_N | 1977–2018 | 0.02 mg/L | 0.28 mg/L | 0.046 mg/L | 0.029 |
| NO$_3$NO$_2$_N | 1968–2018 | 0.02 mg/L | 0.921 mg/L | 0.27 mg/L | 0.201 |
| Temperature | 1968–2018 | | | 18 °C | 4.160 |

The frequency distribution plots (Figure 2) show that most of the Chl−a data are centered near the low-range values, below its average. A similar situation for HN$_4$_N and TP which most of the values fall below 0.1 mg/L and 0.15 mg/L, respectively. Temperature and NO$_3$NO$_2$_N values are distributed stochastically with the most values falling below 20 °C and 0.5 mg/L, respectively. KJEL_N and DO data values have relatively followed the normal distribution and most of the data centered around their averages.

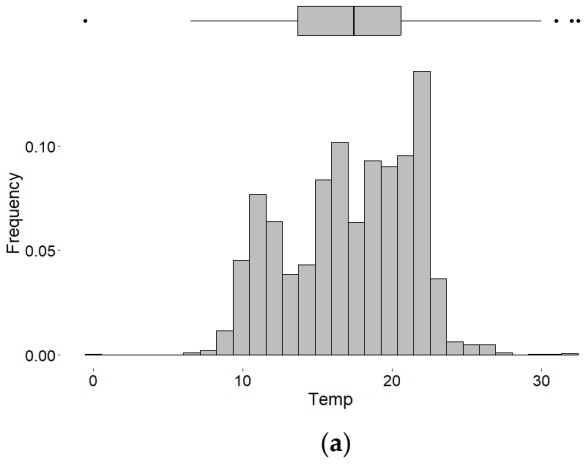

(**a**)

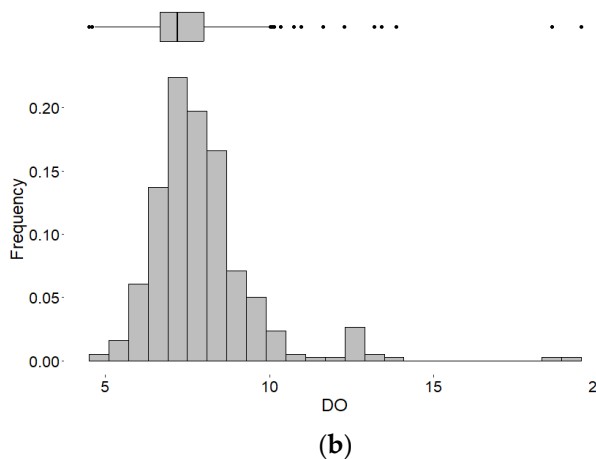

(**b**)

**Figure 2.** *Cont.*

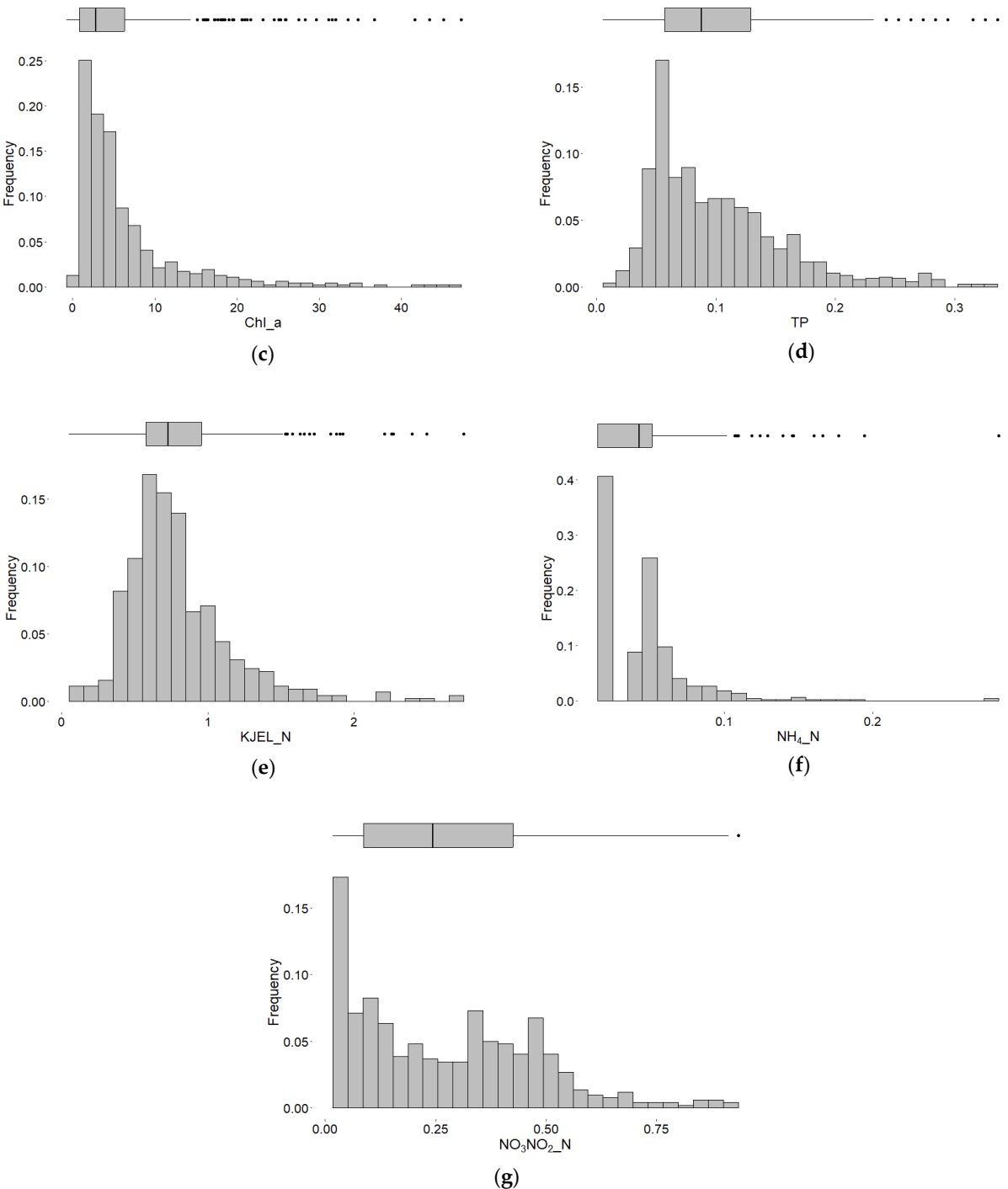

**Figure 2.** Frequency distribution plots of targeted water quality parameters in the Vaal Dam: (**a**) Temperature; (**b**) Dissolved oxygen; (**c**) Chlorophyll-a; (**d**) Total phosphorus; (**e**) Organic nitrogen; (**f**) Ammonia; and (**g**) Nitrate and Nitrite. The extreme values falling outside whiskers of the boxplots represented by black dots.

The time series of the temperature showed seasonal variations which have a strong influence on DO levels, the DO concentrations are decreasing during summer months when the temperature and Chl−a are relatively high (Figures 3 and 4c). The Chl−a concentration time series recorded many peak values (Figure 3) on some erratic dates. The time series plots of the TP and KJEL_N also show some recorded high values corresponding to the same dates of Chl−a high values, reaching up to 1.15 mg/L and 18 mg/L for TP and KJEL_N, respectively, mainly during the period between 2015 and 2017 (Figure 4c) and

between 2004 and 2008 for phosphorus where it jumped up to 0.75 mg/L (Figure 4b). DO, $NO_3NO_2\_N$, and $NH_4\_N$ did not show a clear correlation with the Chl−a time series. But in general, the time series of $NO_3NO_2\_N$ showed low values during the high levels of Chl−a concentrations, and it appears to show a periodic cycle (1.5-year) during the last decade (Figures 3 and 4c).

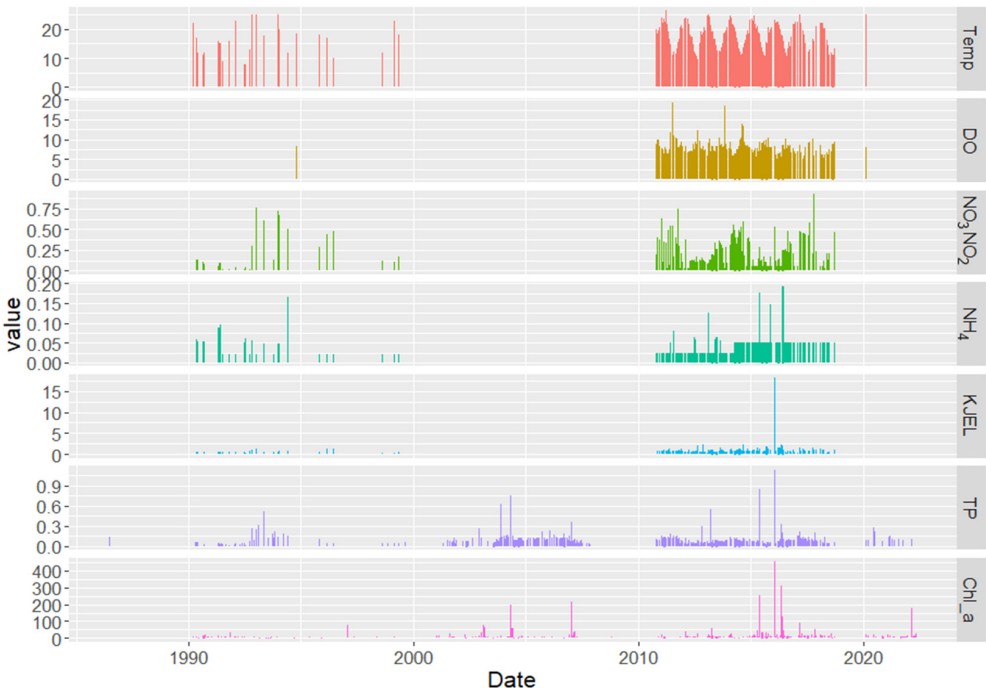

**Figure 3.** Time series of the targeted water quality parameters in the Vaal Dam.

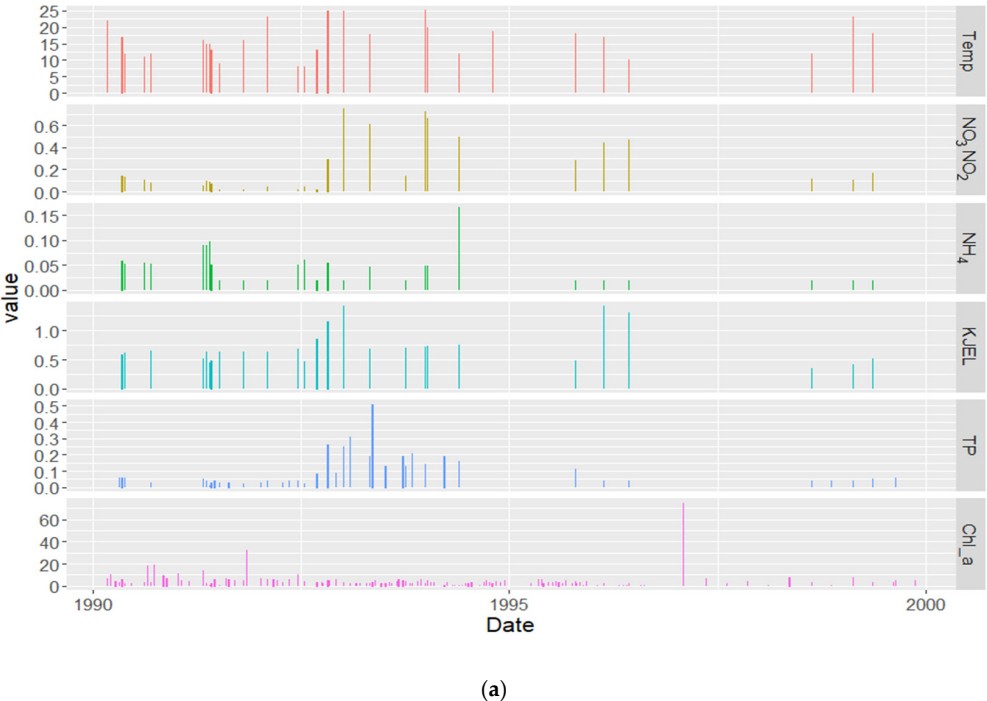

(**a**)

**Figure 4.** *Cont.*

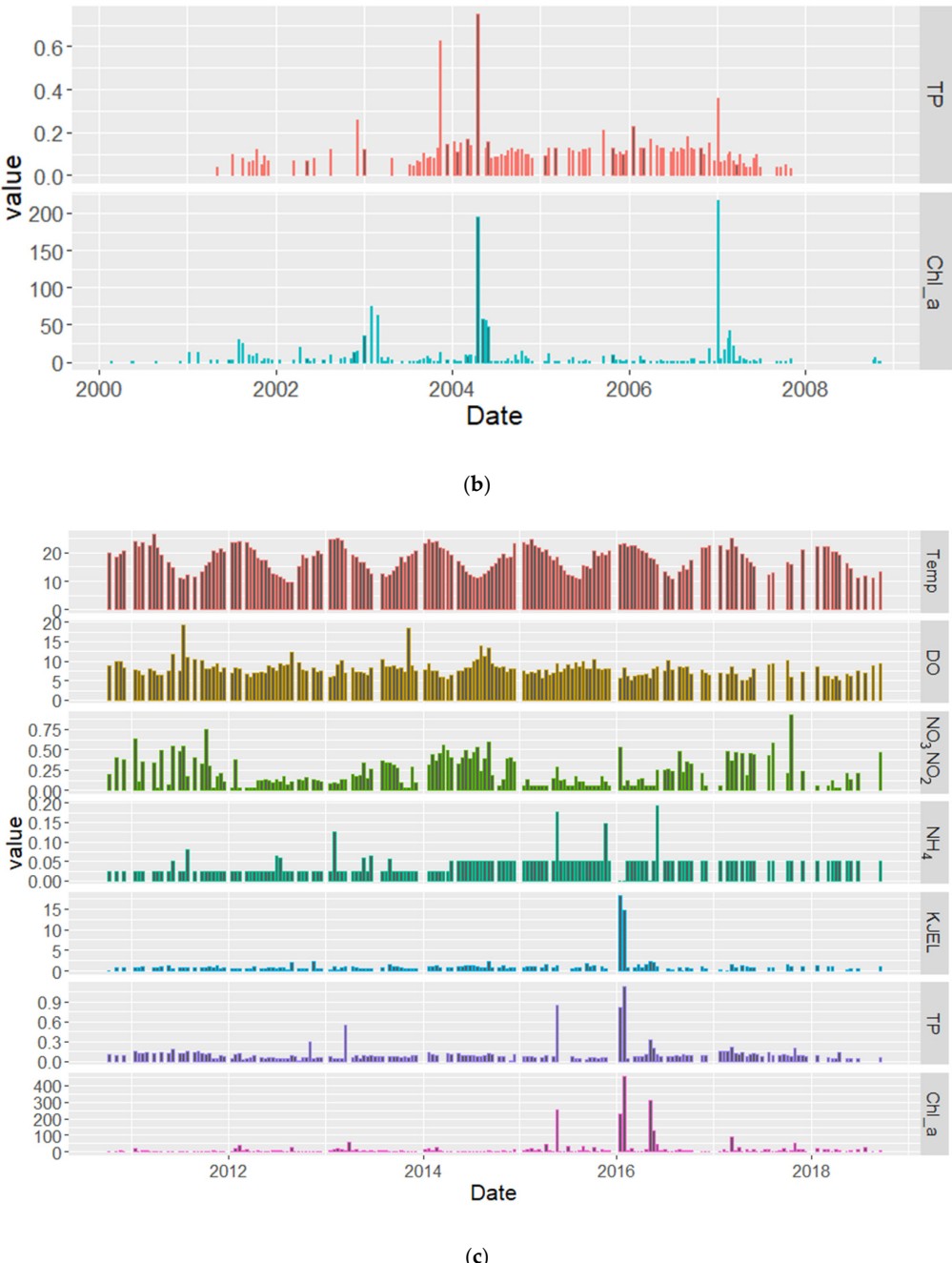

**Figure 4.** Decadal time series of the targeted water quality parameters in the Vaal Dam (**a**) first decade 1990–2000, (**b**) second decade 2000–2010, and (**c**) third decade 2010–2020.

Figure 4 shows the decadal time series for the targeted parameters. In the period between 1990 to 2000, there were very poor records (Figure 4a) which show relatively low values of chlorophyll-a concentrations except on specific dates. TP and KJEL_N show some high values between 1993 and 1995 but there are no correlated peaks with Chl−a during this period due to the poorness of the records and no matchup in their measuring dates. Between 2000 and 2010 only records for Chl−a and TP were available, some high values were recorded on specific dates (Figure 4b). The best records were found between 2010 and 2018 which covered all targeted parameters (Figure 4c), extremely high values were noticed for some of the targeted parameters during this decade. There are increasing decadal averages of Chl−a, TP, and KJEL_N, while there were decreasing decadal averages of $NO_3NO_2\_N$ and no significant change in the $NH_4\_N$ decadal average (Table 2).

**Table 2.** The decadal averages of the studded water quality parameters.

| Decades WQ Parameter | 1st Decade (1990–2000) | 2nd Decade (2000–2010) | 3rd Decade (2010–2020) |
|---|---|---|---|
| Chl−a (µg/L) | 4.75 | 10.51 | 16.7 |
| TP (mg/L) | 0.1043 | 0.1096 | 0.1119 |
| KJEL_N (mg/L) | 0.8 | - | 1.14 |
| $NO_3NO_2$_N (mg/L) | 0.246 | - | 0.225 |
| $NH_4$_N (mg/L) | 0.04 | - | 0.043 |
| DO (mg/L) | - | - | 7.93 |
| Temp. (°C) | 17.9 | - | 18 |

From the decadal trend analysis (Figure 5a–c), Chl−a shows a decreasing trend during the first decade from 1990 to 2000 with low trend average values ranging from ~5.5 to ~4 µg/L and it remains constant during the second decade from 2000 to 2010 with higher mean trend average value, then it showed an upward trend during the last decade from 2010 to 2020, which raised from ~7 to 30 µg/L. TP average trend levels showed the same pattern as Chl−a, it showed a downward trend from around 0.137 mg/L to around 0.075 mg/L during the first decade and showed a nearly constant trend during the second decade with a trend average value of ~0.11 mg/L before it started showing an upward trend during the last decade, where the trend average value increased from around 0.11 to 0.125 mg/L. KJEL_N and $NH_4$_N followed the same pattern as Chl−a and TP, they showed a downward trend during the first decade from ~0.95 mg/L to ~0.65 mg/L and from around 0.053 mg/L to 0.025 mg/L, respectively, and they showed an upward trend during the last decade, KJEL_N from 0.75 to 1.75 mg/L and $NH_4$_N from ~0.028 to 0.06 mg/L. DO showed a decreasing trend during the last decade which decreased from ~9 to ~7 mg/L. The $NO_3NO_2$_N concentration showed an upward average trend during the first decade from ~0.22 to ~0.26 mg/L while remaining nearly constant with an average trend value of around 0.225 mg/L in the last decade. The average trend of temperature shows a slight decrease from ~20 to ~17 °C from 1990 to 2000 and a nearly constant average trend value of ~18 °C from 2010 to 2020.

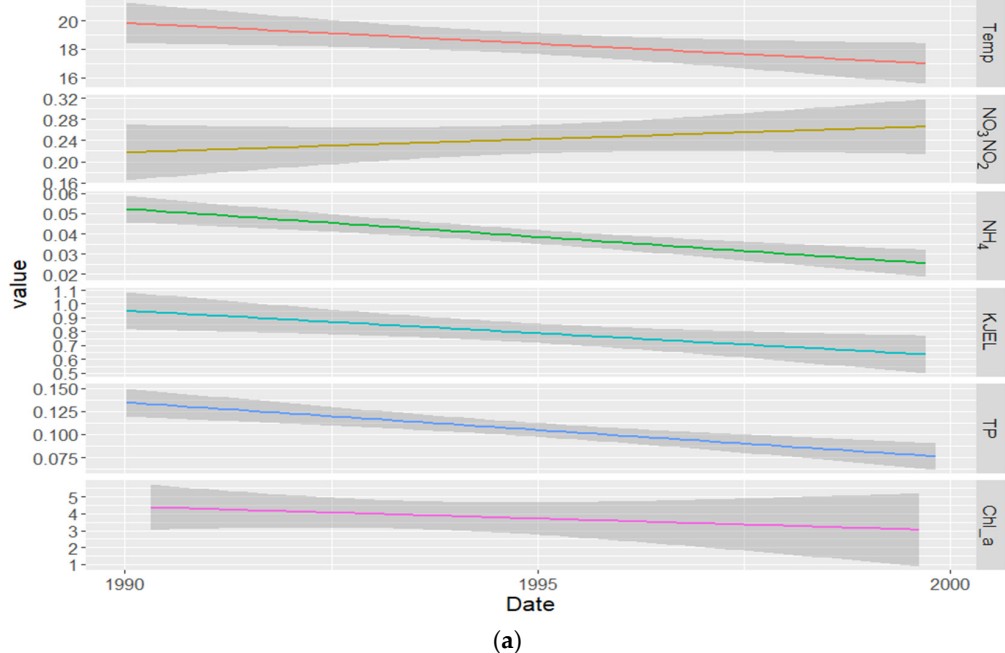

(**a**)

**Figure 5.** *Cont.*

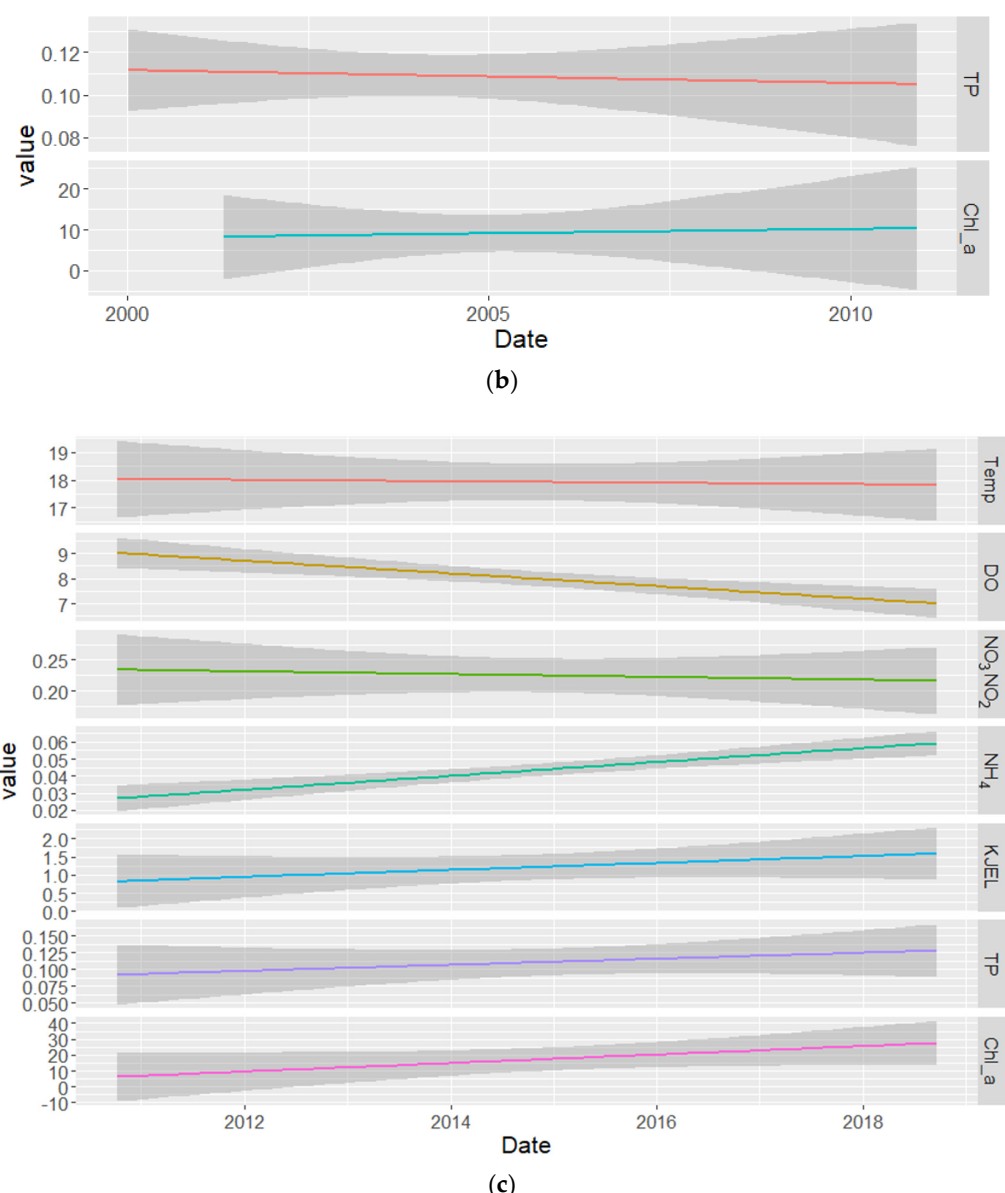

**Figure 5.** Decadal trends of Chl−a, TP, DO, NO$_3$NO$_2$_N, NH$_4$_N, KJEL_N, and Temperature in the Vaal Dam. (**a–c**) showing the decadal trends for the first, second, and third decades respectively.

The regression analysis between the chosen variables showed positive correlations between Chl−a and TP; Chl−a and temperature; Chl−a and KJEL_N, also showed positive correlations between KJEL_N and TP, and KJEL_N and temperature (Figure 6) while s showing negative correlations between Chl−a and DO, Chl−a and NO$_3$NO$_2$_N and between KJEL_N and DO but no correlation between Chl−a and NH$_4$_N (Figure 6).

The satellite-based time series analysis showed the spatial and temporal variability and the HABs dynamics across the dam. The satellite data were normalized to compare the variation between those mentioned on different dates. In general, the images showed high levels of Chl−a concentration which is linked to the summer blooms and the load of nutrients to the system with the summer storm runoff, for example, the image dated 7 February 2016, was captured 10 days after a very high record of Chl−a concentration at the measuring point (452 µg/L). Generally high productivities were seen near the Vaal and Wilge rivers discharge areas in most of the images, also high productivity levels were seen where the reservoir is shallow and meandering (Figure 7).

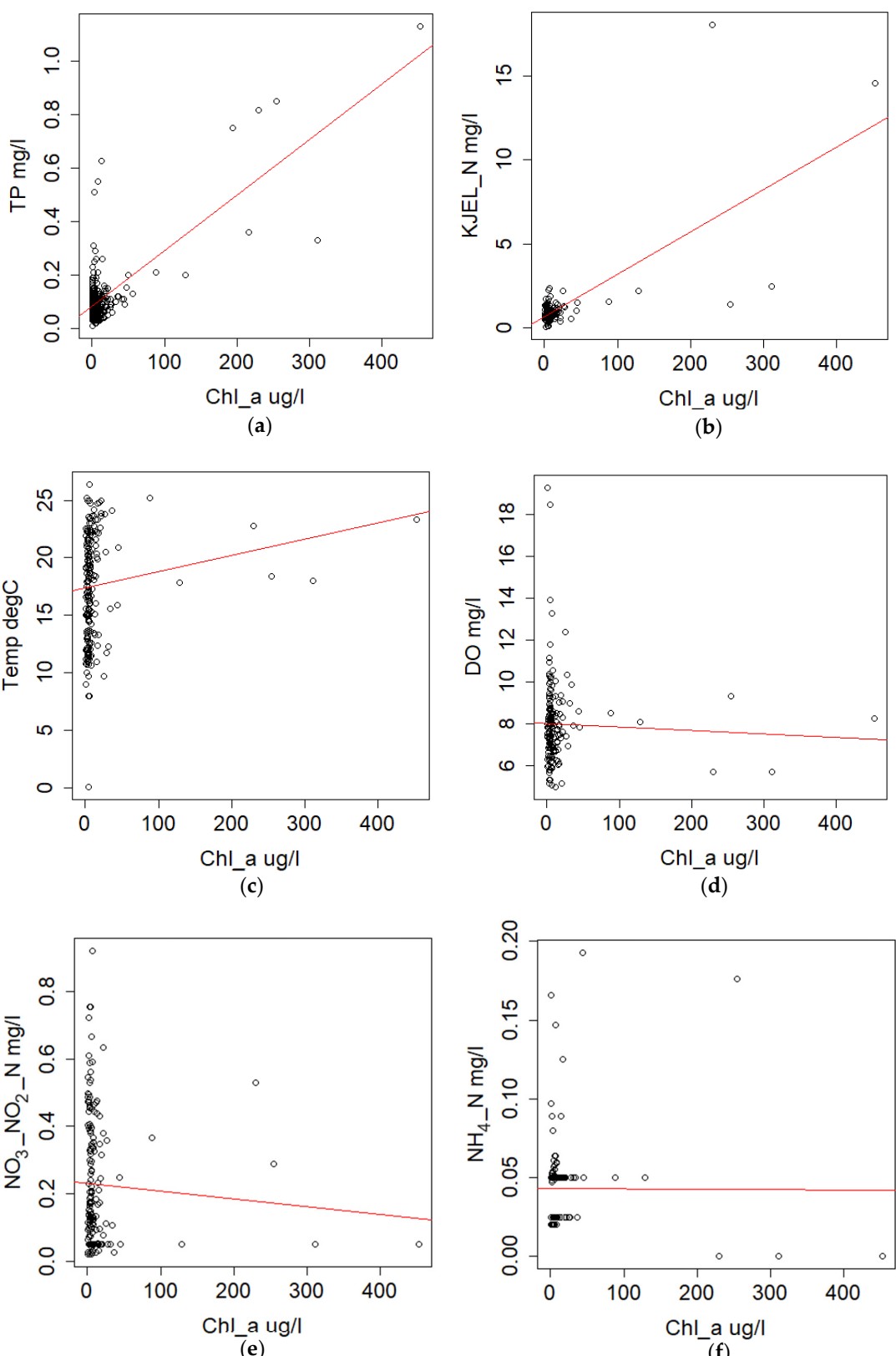

**Figure 6.** Scatter plots between Chl−a and various water quality parameters; (**a**) Chl−a vs. TP; (**b**) Chl−a vs. KJEL_N; (**c**) Chl−a vs. temperature; (**d**) Chl−a vs. DO; (**e**) Chl−a vs. $NO_3NO_2$; and (**f**) Chl−a vs. $NH_4$. The trend lines (red) showing the relationship between the two variables composing the scatter plot.

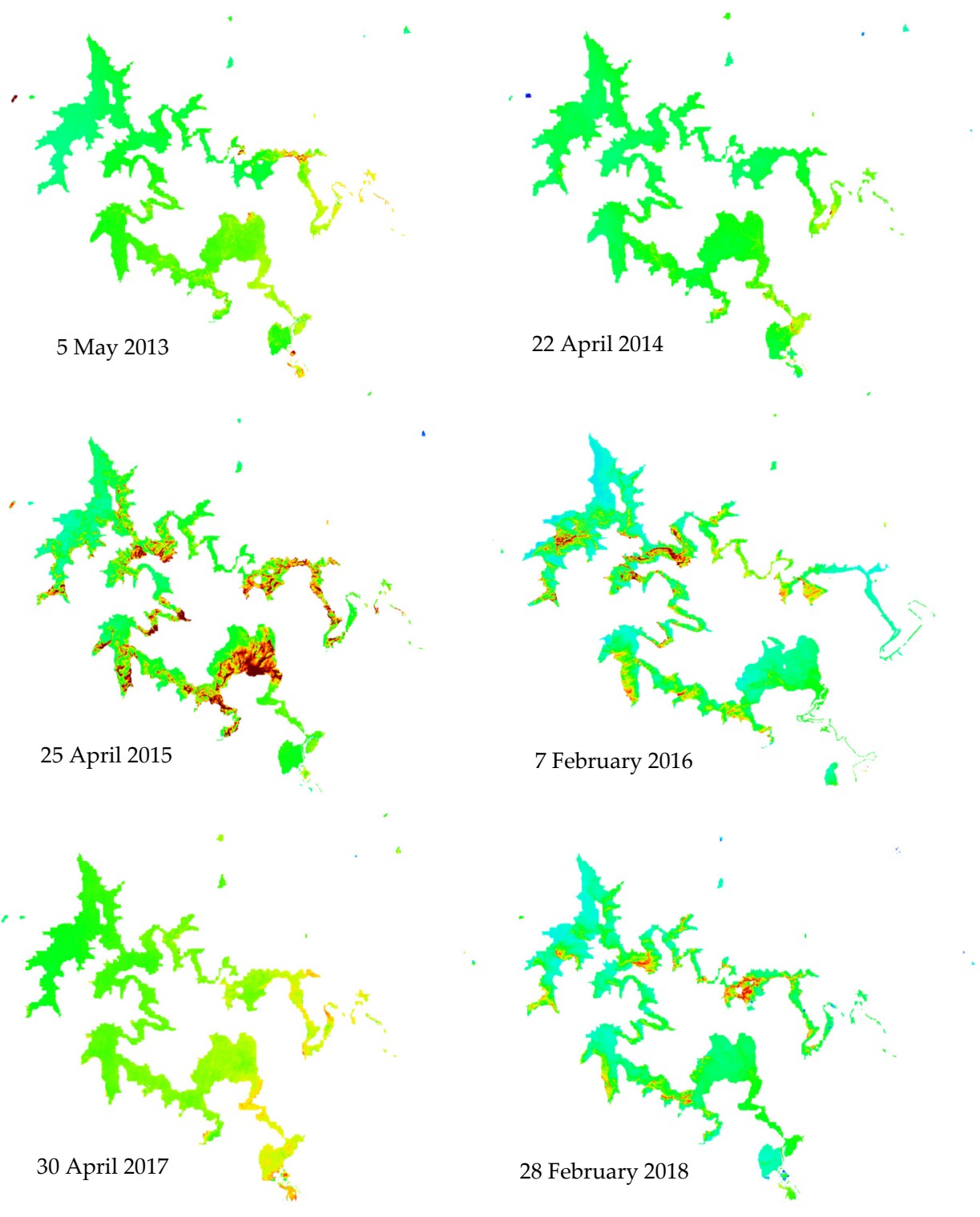

5 May 2013

22 April 2014

25 April 2015

7 February 2016

30 April 2017

28 February 2018

**Figure 7.** *Cont.*

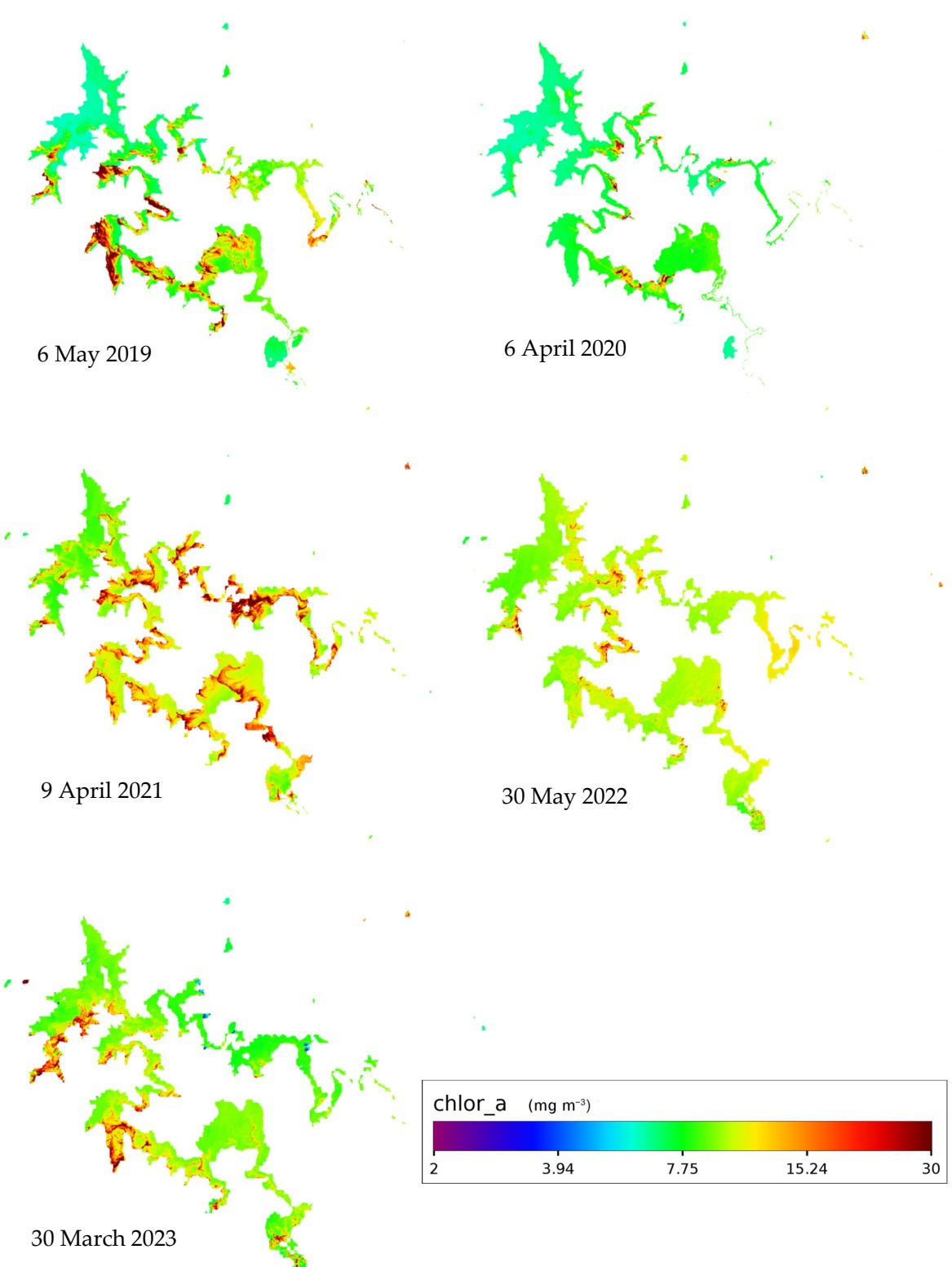

**Figure 7.** Productivity in the Vaal Dam between 2013 and 2023 from Landsat.

## 5. Discussion

The data have been explored to uncover the important water quality properties. The perusal of the graphs reveals some extremely high values as well as some trends, it also reveals the seasonality of temperature and dissolved oxygen. The distribution frequency plot of chlorophyll−a (Figure 2) after excluding the very high values, shows that most of

the data is centered near its lower range, below 10 µg/L. This illustrates that most of the time, the chlorophyll−a concentration in this measurement point remains below its average value. Algal blooms are spatially and temporarily varying, it might happen that an excess bloom might occur at the location of the measuring point during the time of sampling. This might explain the extremely high values of Chl−a concentration. The distribution plots of TP, $NH_4\_N$, and KJEL_N after removing the very high values, also show that most of the data are centered relatively near their lower values in the reservoir water. Therefore, the extremely high to very high values probably indicate that some types of pollution ended up reaching the reservoir which apparently increases the capacity of the reservoir water to support high rates of biomass productivity during such high nutrient load times.

The productivity in the open water systems is a function of multi biophysical factors, such as light, temperature, nutrients, etc. In many aquatic systems, some limiting nutrients (mainly TP and N) are considered the main drivers of HABs [35,36]. However, the time series plot (Figure 3) showed that TP levels between 1990 and 2022 are relatively low except on some specific dates, a corresponding increase in Chl−a and KJEL_N concentrations at such specific dates were detected, which jumped to extremely high values. These are clearer when we look closely at the decadal time series plots in Figure 4b,c. Thus, this explains that the dam water HABs productivity is primarily driven by TP and KJEL_N when their concentrations suddenly jumped to high values. The water temperature time series in Figures 3 and 4a,c shows no significant change, and its decadal trends show a slightly decreasing trend for the first decade (1990 to 2000) in Figure 5a while showing no increasing or decreasing trend during the last decade (2010–2020) in Figure 5c. The average temperatures for the first and last decades were 17.94 and 18.04 °C, respectively, which suggests a slight increase in the average water temperature between the first and last decades. This situation of increasing the decadal average of Chl−a during the study period while the temperature remains with no significant change suggests that the productivity in the Vaal Dam is not limited to the changes in the temperature during the study period, but its value ranges are ideal for algal growth.

Chl−a decadal trends (Figure 5a–c) showed a slight decrease in the first decade (1990 to 2000) and remained constant within the second decade (2000 to 2010) before it increased noticeably during the last decade (2010 to 2020). The average decadal concentrations were 4.75, 10.51, and 16.7 µg/L, respectively. Such an increasing trend of the Chl−a during the last decade alongside the increasing decadal average through the study period raises a significant warning to the situation of algal blooms in the dam. It will become a serious challenge facing the authorities to control this rising impact of HABs in the future. The significant increases in the Chl−a decadal average might be affected by the erratic extremely high values that occurred at individual dates during the second and third decades, for example, when excluding all Chl−a values above 50 µg/L, the decadal averages will decrease to 6.26 and 10.38 µg/L for the second and third decades, respectively. It still has high jumps on the decadal averages through the last three decades.

TP, KJEL_N, and $NH_4\_N$ decadal trends followed the same trend behavior of the Chl−a in the first and last decades, where they were associated with general upward and downward trends of Chl−a. The average decadal concentrations of TP were 0.1043, 0.1096, and 0.1119 for the first, second, and third decades, respectively. These concentrations show that the TP concentrations were low except on the above-mentioned individual dates where they jump to high levels greater than the hypertrophic TP threshold level which is 0.25 mg/L [15]. For KJEL_N, the decadal average concentrations were 0.8033 and 1.1417 mg/L for the first and last decade, respectively, and $NH_4\_N$ were 0.0397 and 0.0431 mg/L, respectively. However, the corresponding behavior of the Chl−a, TP, and KJEL_N decadal trends alongside the erratic high Chl−a values recorded at the same individual dates of very high TP and KJEL_N records, support that they are the driving factors of the algal blooms within the Vaal Dam.

On the other hand, the $NO_3NO_2\_N$ concentrations showed a negative trend compared to the Chl−a decadal trends during the first and last decade and DO also showed a negative

trend during the last decade. The $NO_3NO_2$ decadal average concentrations were 0.2455 and 0.2248 mg/L, respectively, while the average concentration of DO was 7.928 mg/L. The trend of the DO is well understandable because dissolved oxygen is usually consumed during excessive algal blooms which have been known through the last few decades [37], but this situation might not be applicable to the behavior of $NO_3NO_2\_N$ trends. The analysis of the results suggests that it has no direct relation with the Chl−a trend, but a paper [38] focused on sources and forms of the most important nitrogen substrate for blooms in eutrophic Lake Erie suggested that the $NO_3^-$ was the most important source of N except in late blooming stages where phytoplankton relays on recycled N derived from dissolved organic nitrogen. Their study further showed that the $NO^{3-}$ depletion was related to the consumption by phytoplankton during its blooms showing a negative relationship. In this study, $NO_3NO_2\_N$ also showed a negative correlation with Chl−a, but the assumption of consumption by the HABs needs more detailed investigations. The regression between the targeted WQ parameters showed a strong correlation between Chl−a and TP, Chl−a and KJEL_N in great agreement with the trend analysis. It also showed a positive correlation between Chl−a and temperature while there was no correlation between Chl−a and $NH_4\_N$ which explains that $NH_4\_N$ is not a driving factor for HAB in the Vaal Dam. Like the decadal trend plots, the regression showed a negative correlation between the Chl−a, DO and $NO_3NO_2\_N$.

The need for more effective monitoring of the temporal and spatial distribution of HABs in the Vaal Dam Reservoir has led to the use of satellite remote sensing techniques. The satellite data gives a synoptic view of the spatial distribution and the temporal frequency of HABs. Thus, it is an effective tool for the detection, mapping, and monitoring of the blooms. The spatial distribution of HABs in the reservoir has been analyzed using Chl−a concentrations derived from satellite data. Despite the presence of only one ground station for validation, the use of remote sensing data is justified based on the robustness of the employed algorithms. The accuracy of the retrieved data have been confirmed through rigorous evaluation of these algorithms. This assessment included correlating the in-situ data point with 16 distinct satellite images obtained across varying dates, seasons, and weather conditions. From each image, a singular value corresponding to the measuring point coordinates was extracted. The resulting $R^2$ value indicates a reasonable fit, thereby affirming the reliability of the algorithms for chlorophyll retrieval in the Vaal Dam. Additionally, the observed accuracy of the retrieved data underscores the effectiveness of this approach, despite the limited number of in-situ data points utilized for evaluation. Notably, observations indicate temporal variations in Chl−a concentrations, including instances of elevated levels of Chl−a concentrations in 2015 and the following summers. Such high concentrations are strongly correlated with the in-situ measured values of Chl−a, TP, and KJEL_N during April and May 2015, and January 2016. High productivity was also noticed in images of 2019, 2021, 2022, and 2023. These high productivities may relate to the periods where TP and KJEL_N levels increase due to the increase in human activities such as the discharge of partially treated or untreated wastewater, or through runoff from urban and agricultural areas around the dam. For example, the media reported some waste found its way to the reservoir on 23 July 2015, a report mentioned that an uncontrolled sewage discharge overwhelmed the Deneysville town which is located on the Vaal Dam bank, just next to the dams wall (https://mg.co.za/article/2015-07-23-sewage-in-gautengs-drinking-water/), accessed 7 March 2023.

The satellite data showed high productivity in the Vaal Dam for the period from 2019 up to 2023 while there are no historical records available on the website during this time, which put some warnings of increasing nutrient loading to the dam reservoir recently. The geometry of the dam and the fluxes of the Vaal and Wilge rivers draining into it are directly linked to the pattern of HAB spatial dynamics, the shallow parts of the dam as well as the meandering areas where the velocity of the water decreases always exhibit relatively high blooms compared with the rest of the dam. Satellite monitoring is useful in detecting the frequency of the blooms. In this study, we processed one image each summer during the

last decade, but there is a potential to monitor the spatial and temporal dynamic of the blooms every 16 days using Landsat−8 images only, or shorter intervals if using a harmony of Landsat-8 and Sentinel−2 a and b satellite data.

## 6. Conclusions

This study revealed that the Vaal Dam productivity is a function of TP and KJEL_N levels. The analyzed data between 1986 and 2018 showed a positive correlation of TP and KJEL_N with chlorophyll-a concentrations. They are the HABs drivers alongside the seasonal effect of surface water temperatures which enhances the productivity levels. When TP and KJEL_N levels are low, the Chl−a concentrations are usually below the threshold level of the eutrophic conditions of 7 µg/L. Chlorophyll-a retrieved from Landsat data showed the temporal and spatial dynamics of HABs over the past decade. Time series analysis of the data has shown the effect of sudden rises of the driving parameters on productivity levels, the productivity increases with an increased flux of TP and KJEL_N. The relatively high concentrations were observed where the reservoir is shallow and meandering and where restricted water circulation occurs due to the low level of mixing and near Vaal and Wilge rivers discharge points. According to the location of the in-situ measurement point, which is in the Vaal River stream near the dam wall, the measured Chl−a concentration might be much less than the concentrations on many other parts of the dam reservoir because the HABs can be washed out by high flow speed. Thus, one measuring point for this large dam is not enough to reveal the overall situation of the chlorophyll-a concentration on the dam, mainly when considering the spatial variations. Therefore, more water quality monitoring points are needed and to be well distributed among the reservoir for a better understanding of the spatial variations and to give valuable information to be used for satellite monitoring algorithm building and adjustments. However, we recognize the inherent limitations of relying on a solitary sample point. Therefore, to enhance the reliability and generalizability of our findings, we emphasize that this approach serves as an initial exploration to provide the general trend of the water quality. However, it also highlights the need for further investigation and data collection to strengthen the robustness of our findings. The results obtained from this time series analysis revealed a high level of anthropogenic impacts on the Vaal Dam which is being used to provide drinking water for Gauteng province, and for agricultural and industrial uses in the region. The periodic deterioration of the quality of the dam water over some stochastic dates was aggravated by the discharge of nutrient-rich, poor-quality effluents from the wastewater treatment works in the dam catchment.

**Author Contributions:** Conceptualization, A.A.O.; Data curation, A.A.O.; Formal analysis, A.A.O.; Investigation, A.A.O.; Methodology, A.A.O.; Resources, A.A.O.; Software, A.A.O.; Supervision, E.M.A., K.A.A. and T.A.A.; Validation, A.A.O., E.M.A., K.A.A. and T.A.A.; Visualization, A.A.O., E.M.A., K.A.A. and T.A.A.; Writing—original draft, A.A.O.; Writing—review and editing, A.A.O., E.M.A., K.A.A. and T.A.A. All authors have read and agreed to the published version of the manuscript.

**Funding:** This research received no external funding.

**Data Availability Statement:** The data presented in this study are publicly accessible in (http://www.dwa.gov.za/iwqs/wms/data/WMA08_reg_WMS_nobor.htm, accessed on 2 February 2023).

**Acknowledgments:** The authors would like to show their acknowledge and appreciation to the reviewers and the editor for their valuable comments and suggestions.

**Conflicts of Interest:** The authors declare no conflict of interest.

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
