# Peer review of "Time Series Analysis of Water Quality Factors Enhancing Harmful Algal Blooms (HABs): A Study Integrating In-Situ and Satellite Data, Vaal Dam, South Africa"

_water, doi:10.3390/w16050764_

Round 1

Reviewer 1 Report

Comments and Suggestions for Authors

Time series analysis of water quality factors enhancing harmful algal blooms (HABs): a study integrating in-situ and satellite data, Vaal Dam, South Africa

Recommendation: Major Revision

The study aimed to investigate the relationship between HABs, and the potential nutrients and environmental processes during the times of peak blooms within the last few decades using historical water quality records from the Vaal dam reservoir. Review comments must be addressed first before accepting this manuscript. 

Comments:

1)     The authors should elaborate more on the specific analyses conducted using R v4.3. Include all statistical methods used for calculating long-term averages and generating different types of plots such as time series plots, frequency scatter plots, and decadal trend plots.

2)     I suggest that the authors provide references or links to code/scripts used in the analysis to ensure reproducibility and transparency of the study.

3)     I suggest that the authors mention ethical considerations on the use of satellite imagery or historical data to ensure compliance with data usage regulations.

4)     Are there gaps in the historical water quality data? How did the authors handle the incomplete or missing data in their analysis?

5)     Table 1 is informative but it will be helpful to include standard deviation or other measures of variability in the obtained values.

6)     I suggest that the authors add more discussion in the spatial distribution of HABs and patterns that can be observed over time. Also, give emphasis on the importance of satellite data in elucidating the HAB bloom dynamics of the reservoir.

7)     I suggest that the authors properly explain the effects of outliers (extreme values) in their analysis. What is the reason behind these extreme values as well as its impact on the overall presentation of their statistical analysis.

8)     In Figure 5, is it possible to include statistical measures of trend significance? Additional discussion is needed about these trends on water quality and its potential impact in the Vaal Dam.

Comments on the Quality of English Language

Minor editing of English language is required. 

Author Response

Dear Reviewer 1

Kind regards

Reviewer 2 Report

Comments and Suggestions for Authors

This paper studies the work of monitoring harmful algal blooms (HABs) using in-situ observation data and remote sensing images. Time series analysis of  water quality factors of the study area is made. The correlations between different factors are revealed and the distribution trend of HABs in different years are shown. My suggestions to improve the quality of this paper are as follows:

1) The language presentation of the manuscript is poor in some places, and gramma errors can be found in many places. Please extensively improve the language presentation.

2) Please delete the template contents, "0. How to Use This Template. The template details the sections........"

3) The abstract is too lengthy. The authors should point out the major work they have done and the major conclusion they have drawn.

4) The introduction section is also unsatisfactory. The authors have mentioned too much background information about the study area, but rarely mentioned the research situation about monitoring HABs, especially, the monitoring of HABs using remote sensing technique.

5) The data from only one sample point is acquired to reveal the situation of water quality of the study area. Is it enough for only one point to represent the situation of the study area?

6) Line 224, "A simple two-band ratio of blue-green based ocean color algorithm was applied [38] for Vaal Dam water modeling productivity". The authors should clearly describe in the paper how to generate the Chl-a concentration data using the remote sensing images. This is one of the key issues of this study. 

7) Does the  Chl-a concentration generated by remote sensing data meets the observation results from the in-situ station? or to say, how about the accuracy of the Chl-a concentration generated by remote sensing data? The authors only mentioned that the generated data has the similar trends with the in-situ observed data, but that is not enough.

Comments on the Quality of English Language

Extensive editing of English language required.

Author Response

Dear Reviewer 2

Kind regards

Reviewer 3 Report

Comments and Suggestions for Authors

The paper focuses on investigating the relationship between HABs and various water quality parameters in the Vaal Dam, South Africa. It uses in-situ and satellite data to analyze the impact of nutrients like phosphorus and nitrogen on HABs. The study also examines spatial and temporal dynamics of HABs using Landsat-8 and Sentinel-2 satellite images. Key findings include the positive correlation between total phosphorus, organic nitrogen levels, and chlorophyll-a concentration, indicative of HABs. The study further discusses the impact of dam geometry and river discharges on HABs spatial distribution. Generally, this paper is clear written. But there are some minor issues which should be fixed before publication.

1. The authors should state the significance of the paper in abstract and introduction.

Comments on the Quality of English Language

2. There are some typo, for example, “humanbeing” in line 54, “scutter plots” in line 19.

3. The NO3NO2_N is not standard symbol.

Author Response

Dear Reviewer 3

Kind regards

Reviewer 4 Report

Comments and Suggestions for Authors

This manuscript describes the temporal trends in Chl-a and other parameters in the Vaal Dam Reservoir in South Africa.  Approximately 30 years of data are analyzed.  The authors suggest that Chl-a levels have greatly increased over time, and that this increase is largely driven increases in total phosphorus, ammonium, and Kjeldahl nitrogen (TP, NH4_N and KJEL_N in the manuscript).  They also have provided some yearly snapshots o algal blooms based on remote sensing data.

 Comments

1) line 440 – “The results obtained from this time series analysis revealed a very high level of anthropogenic impacts on the Vaal Dam which is being used to provide drinking water for the Gauteng province, and for agricultural and industrial uses in the region.”

I would argue that it is clear that Chl-a levels have increased (Table 2), but the mechanism leading to the increase in Chl-a is not clear.  TP and KJEL_N have increased only slightly since 1990 (~10-20%), yet Chl-a has increased almost four-fold.  The authors attribute this to anthropogenic activities, but what are these activities? 

The data in Table 2 are very confusing.  If TP, NH4_N and KJEL_N show only moderate increases, and if N and P are considered to the most important limiting nutrients (there is a lot of literature support for that), what caused the four-fold increase in Chl-a?

The authors make the point that nutrients may sometimes increase due to e.g. sewage overflows.  That makes sense, but the overflows are transient.  On the other hand, there is no mention of the water residence time in the Vaal Dam Reservoir.  If the residence time is long, that would perhaps lead to elevated N and P for a period of time.  But the elevations in N and P appear to be brief (Fig. 4), however, there are spikes of Chl-a associated with elevated N or P.   

But if the average (long-term) Chl-a has increased four-fold, there must be an explanation.  Alternatively, I wonder if the average Chl-a has been overestimated in Table 2?  The spikes in Chl-a shown in Fig. 4 could lead to a higher average Chl-a for that decade.  However, the baseline Chl-a may not have greatly increased over 30 years?  That is, there are spikes in Chl-a, but the baseline levels may not have changed very much.  Is that a possibility?

2) line 363 – “The low concentration of Chl-a between 1990 and 2000 may be because of the implementation of the bioremediation project between (the late 1980’s to the early 1990’s) which reduced the Chl-a levels effectively [41].”

What sort of bioremediation project was undertaken?  Did the project remove N and P?  If yes, why is that not evident in the decadal data?  Some additional details are needed for the reader.

3) The authors suggest that the Vaal Dam reservoir has suffered from harmful algal blooms (HABs).  What is the evidence that HABs occurred?  Elevated Chl-a levels suggest a bloom, but that does not necessarily suggest a HAB.  I have a number of related comments regarding the nature of the blooms.

line 125 – “The model showed a promising result of estimating Microcystis sp. in 7-days in advance [25].”

There is a reference cited (25) that mentions the cyanobacterium Microcystis.  However, even in reference 25, there is not a description of other species that may be present.  Reference 27 mentions that both Anabaena and Microcystis are present in blooms in the Vaal Dam Reservoir.  Are the blooms typically predominantly Microcystis in the Vaal Dam Reservoir?  In this manuscript, there is a notable lack of background information about the blooms and their composition.  If the authors are going to suggest that high Chl-a = high Microcystis or high cyanobacterial abundance, there should be some evidence provided.  Otherwise, high Chl-a is simply an indicator of high phytoplankton biomass.

In terms of the phytoplankton composition of water in the Vaal Dam Reservoir, I find it odd that Chinyama et al. Water SA Vol. 42 No. 3 July 2016 is not cited.  These authors actually examined the phytoplankton composition in the reservoir over the course of a year.  This would be useful background information (they mention that Anabaena and Microcystis are the most abundant cyanobacteria, but also mention that at certain times of the year non-cyanobacteria are the dominant phytoplankton).  This relates to my comment that elevated Chl-a indicates a bloom, but not necessarily a HAB.

4) The remote sensing data in the manuscript are snapshots of the reservoir (one snapshot per year).  It would be useful to know if other snapshots (not addressed in the manuscript) also suggested that phytoplankton blooms had occurred.  Alternatively, were phytoplankton blooms only detected in the presented snapshots?

Minor comments

line 133 – The meaning of Case 1 and Case 2 is never explained.  Outside of the South Africa context, these terms likely will mean nothing to most readers.

Author Response

Dear Reviewer 4

Please see the attached

Kind regards

Round 2

Reviewer 1 Report

Comments and Suggestions for Authors

The author revised the manuscript addressing the major comments and suggestions of the reviewer.  I found some of the description and explanation of the paper to be already detailed and complete.  Also, the novel point in this study was already emphasized in the paper.

Please double check the manuscript again for grammatical errors. I still saw some corrections in the revised manuscript. 

Comments on the Quality of English Language

The author revised the manuscript addressing the major comments and suggestions of the reviewer.  I found some of the description and explanation of the paper to be already detailed and complete.  Also, the novel point in this study was already emphasized in the paper.

Please double check the manuscript again for grammatical errors. I still saw some corrections in the revised manuscript. 

Author Response

Thank you very much for your valuable review comments, Please see the attachment.

Reviewer 2 Report

Comments and Suggestions for Authors

The authors have answered some of my question just in the file of "response to comments", but I suggest the authors to point out some issues in the manuscript. For example, "The data from only one sample point is acquired to reveal the situation of water quality of the study area. Is it enough for only one point to represent the situation of the study area?"  "Does the Chl-a concentration generated by remote sensing data meets the observation results from the in-situ station? or to say, how about the accuracy of the Chl-a concentration generated by remote sensing data? The authors only mentioned that the generated data has the similar trends with the in-situ observed data, but that is not enough."  

Author Response

Thank you very much for your valuable comments, please see the attachment.

Reviewer 4 Report

Comments and Suggestions for Authors

I believe that the revised manuscript is improved compared to the original, and I have no further comments.

Author Response

Thank you very much for your valuable comments.